# Research on Sea Spray Distribution of Marine Vessels Based on SPH-FEM Coupling Numerical Simulation Method

**Jiajing Chen** , **Xu Bai \***, **Jialu Wang, Guanyu Chen and Tao Zhang**

School of Naval Architecture & Ocean Engineering, Jiangsu University of Science and Technology, Zhenjiang 212003, China
* Correspondence: baixu@just.edu.cn

**Abstract:** Due to the effect of the maritime environment and low temperature factor, ice phenomena are easily produced while a ship is sailing in a polar location. Types of ice accumulation include sea spray icing, which accounts for 90% of all ice accumulation, and, therefore, sea spray generation is a crucial step in ice accumulation prediction research. In order to investigate the phenomenon of ice formation on ships, this paper uses the SPH-FEM (Smooth Particle Hydrodynamics-Finite Element) coupling approach in this paper, and tracks the data pertaining to the wave current particles by simulating the impact of a single wave on the ship hull under the wave height standard of various sea conditions. Following the numerical simulation, it was discovered that when the sea state reaches five levels, the waves will produce marine sea spray on the simulated hull and distribute them in an arc on both sides of the hull; the phenomenon is influenced by the sea state level, meaning that the arc becomes more obvious as the sea state level rises. Furthermore, the number of sea spray particles formed in different sea state levels accounts for about 9–13% of the range of individual waves, and the rest of the sea spray particles will flow back to the surface quickly after passing through the hull.

**Keywords:** ship ice accumulation; sea spray; marine structures; SPH-FEM coupling method

## 1. Introduction

The impact of low temperature elements and the marine environment causes ice formation in hull structures easily when ships are cruising in polar waters. High levels of ice buildup on the hull structure can result in the obstruction of different exposed systems for ventilation and equipment, as well as safety risks including ship capsizes and fatalities of crew members [1]. Therefore, the primary focus and direction of this field's research has been on the issues linked to hull freezing and ice accumulation, in order to assure the safety of ship navigation and operation. The subject of marine icing forecasting is more complex and depends on many variables, such as ice accumulation will depend on meteorological conditions (wind speed, air temperature, water temperature, and seawater freezing point), the wave height generated, and the vessel speed and motion relative to the wave. The amount of spray, its density, duration, and frequency depend on the ship design features, speed and heading with respect to the waves. In typical polar locations, there are two types of ship hull icing, those being atmospheric icing and sea spray icing [2]. The most influential factor of atmospheric icing depends on meteorological conditions, and the most influential factor of sea spray icing depends on the state of motion of the waves and the ship. The findings of similar investigations indicate that wave-induced marine sea spray account for 90% of the water sources that cause the icing of marine hull structures [3]. The process of oceanic sea spray icing includes sea spray generation, air flow, and heat transfer [4], among which sea spray generation is crucial to the study of pre-ice prediction. There is a dearth of physical modeling and flux measurement of sea spray sources, and the majority of current research on sea spray formation is based on the empirical equation of droplet flux [5,6]. Therefore, further research on the process of sea spray generation is of great significance for icing forecasting.

In recent years, scientists have created a variety of physical models that anticipate the sea spray icing which results from "wave-ship" interactions (Lozowski et al., 2000; Anton Kulyakhtin, 2012; Samuelsen et al., 2017 [7–9]). Most of these models do not include the process of ocean sea spray generation, which includes the procedure of wave impact on the hull and the distribution of sea spray on the hull after breaking up to form sea spray, instead directly combine the droplet flux of the empirical equation with the heat transfer icing process. Despite Lozowski's [7] more recent research findings suggesting that the generation of sea spray can be predicted by using linear wave theory and potential flow theory in a constant uniform wind field and suggesting that the ice accretion rate is determined by the sea spray flux and heat conduction on the surface of the structure, this ignores the actual process of sea spray generation, so the model does not describe the real hull ice accumulation behavior. Anton Kulyakhtin [8] proposed the ice growth model-MARICE after ignoring the effect of thermodynamics in the interaction between sea spray and simulating the flow of sea spray in relation to the surrounding environment of the ship, combined with the icing process of heat transfer, but the droplet flux of this model is obtained from empirical equations, so there is still some room for the research of this icing model on the generation of sea spray and the problem of heat transfer between ice layers. The MINICOG model proposed by Samuelsen [9] is modified by two different empirical formulas of droplet flux to propose an empirical formula based on the relationship between local wind speed and effective wave height and wave period, but the model is affected by the actual sea spray generation process and flux problems, and has certain conditions and limitations.

In general, most ocean icing models have historically relied on empirical equations for ocean droplet fluxes due to the limitations of theoretical development and observational tools [10–12]. This lack of physical modeling of sea spray sources, measurements of sea spray generation and actual fluxes, eventually resulted in discrepancies between icing predictions and actual conditions [13]. The droplet trajectory in the empirical equation depends on gravity, wind speed, initial droplet size and velocity, and many other parameters. In the past studies, droplet concentration was usually ignored because droplets were assumed to be diluted and moved individually over the container without interacting with each other. Individual droplets experience more drag during their flight and therefore require high initial velocities to overcome gravity and drag. The air flow field influenced by the surrounding droplets has less drag on a particular droplet and, therefore, the droplet travels at an increased speed and maximum height. This suggests that a cloud of droplets traveling as droplets or filaments can reach higher heights with lower jet velocities [3]. Therefore, the theoretical model is currently limited to predicting mono-droplet trajectories, but CFD can simulate numerous droplets in a single domain. Therefore, one of the key components of current research on icing phenomena in maritime structures and a key factor in improving icing forecasts is the investigation of physical modeling and flux measurements of the sea spray formation process [14,15]. Regarding sea spray sources, sea spray generation can be divided into a variety of stages, such as free surface modeling, wave thumping on the bow, air entrainment during impact, formation of water film and jets on the wall after wave thumping, wind-induced water film and sea spray fragmentation, and sea spray trajectory in contact with the ship's deck surface, and others [16–18]. Modern mainstream research on icing of marine structures has mainly focused on the formation of sea spray after wave impact on marine structures, and the research has achieved certain results, but there are also some controversies and shortcomings.

Horjen and others [19–21] optimized the procedure used to simulate sea spray generated by wave impact based on the RIGICE and MARICE model and derived a new equation for liquid water content, but their observations only matched their own data. In recent years, many scholars have begun to observe sea spray through experimental modeling, and many related models based on computational fluid dynamics (CFD) have been developed. Amir Banari et al. [22,23] combined the VOF (Volume of fluid) method to propose the trajectory of the sea spray form after wave thumping when it comes into contact with the hull,

which can visualize the changing form of the sea spray, but cannot predict its droplet flux. Kees et al. [24,25] introduced a new Level-Set method to solve the robustness of large-scale complex free-surface deformation. However, after all, the simulation of airflow entrainment is very important in the study of wave-hull impact problems [26], so the method also has limitations and do not apply to morphologies where the air is intercepted, or where the liquid contains gas; Greco and Lugni et al. [27] developed a numerical and experimental method to estimate various statistical analyses of the performance of wave lapping on the deck, and the relevant parameters such as pressure and collision position on the deck were obtained from the sea spray morphology. The shortcoming of this method is the lack of microscopic application of the rupture mechanism; After this, Dehghani et al. [28] developed a new three-dimensional model to simulate wave impact on MFV ships using droplet size and velocity characteristics in an attempt to derive the droplet size and velocity distribution of sea spray after wave impact on the hull from a microscopic perspective. In addition, Shipilova et al. [29] used the LWC model of Zakrzewski [30] to generate sea spray in front of two types of ships and assumed that the sea spray was generated in the form of squares and used the Eulerian-Lagrangian method into sea spray simulation, but this method requires setting the sea spray duration by itself and cannot simulate the periodic motion of the sea spray. Finally, due to the computational cost, most Lagrangian methods cannot be used for sharp deformations of the free boundary at large scales.

The present study is based on the idea proposed by Anton Kulyakhtin [31] and other scholars that the phenomenon of ice accretion on marine structures is caused by the periodic superposition of marine sea spray formed by waves impacting on the hull of a ship after icing on the hull. Considerable progress has been made in recent years in simulating ocean sea spray and icing phenomena using numerical methods, but none of them have studied the periodicity of sea spray superposition [32]. The droplet flux includes droplet collection and collision coefficient, droplet velocity, liquid water content, frequency and duration of droplet occurrence, of which the last three basic parameters are the key ones to be estimated. In this study, the effect of the wind speed, heave and pitch of the ship is neglected to simplify the problem. In order to address this issue, this paper analysis first makes the assumption that the ship moves in a regular wave and that the outcomes of the two periodic periods are unrelated, reducing the ship motion to the degree of freedom of forward translation only. Second, the periodic superposition problem is divided into smaller, more independent parts for the research based on the multi-scale character and intricacy of the sea spray problem. This study examines the distribution of marine sea spray on a ship's hull caused by a single cycle of regular waves striking the hull in a variety of sea states. The process of wave impact on ship hull is numerically simulated using SPH-FEM (Smooth Particle Hydrodynamics-Finite Element Method), taking into account the viscous effect and wave-hull interaction effect. A new method of ocean sea spray improvement and the distribution probability of sea spray on the hull are established based on the probability distribution law of single period sea spray on the hull. This establishes the groundwork for later research on the periodicity of ice accretion brought on by ocean sea sprays.

## 2. Theoretical Methods

### 2.1. Governing Equations

This paper studies the phenomenon of sea spray caused by the interaction of wave and ship structure phase, which does not involve the subsequent steps of icing proliferation. Therefore, the influencing factors of wind speed and heat transfer are temporarily ignored. In order to detect the proportion of a single period of sea spray particles on the hull, the SPH-FEM coupling method is used in this paper. SPH particles are used as fluid particles of waves, and the hull structure of the phase action is divided into FEM grids [33]. The coupling contact algorithm is added to achieve the effect of phase action. This method effectively improves the efficiency of operation [34]. In the following, wave particles are regarded as water fluid, and the N-S equations describing the mass conservation and momentum conservation of fluid motion are used:

$$\frac{D\rho}{Dt} = -\rho \nabla \cdot \mathbf{u} \tag{1}$$

$$\frac{D\mathbf{u}_w}{Dt} = -\frac{\nabla p}{\rho} + v\nabla^2 \mathbf{u} + \mathbf{g} \tag{2}$$

In the formula: $\mathbf{u}$ is the velocity of fluid particles, $p$ is inter-fluid pressure, $\rho$ is the density of water fluid and $v$ is the kinematic viscosity coefficient of water. After the correction of the Riemann solver, particle-to-particle contact algorithm and the CSPM method [35], the SPH particle approximate expression of Equations (1) and (2) is:

$$\frac{d\rho_i}{dt} = \frac{2\rho_i \sum\limits_{j=1}^{N} \frac{m_j}{p_j}(u_i^R - u_{ij}^R)\cdot\nabla_i W_{ij}}{\sum\limits_{j=i} N(r_{j-r_i}) \otimes \nabla_i W_{ij}\frac{m_{ij}}{\rho_j}} \tag{3}$$

$$\frac{D\mathbf{u}_i}{Dt} = \frac{-\sum\limits_{j=1}^{N} m_j \frac{p_{ij}^*}{\rho_i\rho_j}\cdot\frac{\partial W_{ij}}{\partial x_i}}{\sum\limits_{j=i} N(r_{j-r_i}) \otimes \nabla_i W_{ij}\frac{m_{ij}}{\rho_j}} + \sum\limits_{j=1}^{N} m_j\left(\frac{4vr_{ij}\mathbf{u}_{ij}}{(\rho_i + \rho_j)|\mathbf{r}_{ij}|^2}\right)\frac{\partial W_{ij}}{\partial x_i} + \mathbf{g} \tag{4}$$

where: $W_{ij} = \alpha_D(1 - \frac{q}{2})^4(2q + 1), 0 \le q \le 2; \alpha_D = \frac{7}{4\pi h^2}, q = \frac{|r_i - r_j|}{h}$

Of which, the normal velocity $U_{ij}^R$ and pressure $P_{ij}^*$ on the particle discontinuity surface are:

$$P_{ij}^* = \frac{p_j\rho_i c_i + p_i\rho_j c_j + \rho_i c_i \rho_j c_j(u_i^R - u_j^R)}{\rho_i c + \rho_j c_j} \tag{5}$$

$$U_{ij}^R = \frac{u_i^R\rho_i c_i + u_j^R\rho_j c_j - (p_j + p_i)}{\rho_i c + \rho_j c_j} \tag{6}$$

The parameters of the left and right sides of the discontinuity are $u_i^R$, $p_i$, $\rho_i$, and $U_j^R$, $p_j$, $\rho_j$, respectively. This discontinuity propagates at sound speed $c_i$ and $c_j$ to the left and right sides. The pressure $p$ in formula (5) selects the equation of state used by Monahan and Kos to calculate the free surface flow problem [36]:

$$p = B\left[\left(\frac{\rho}{\rho_0}\right)^\gamma - 1\right] \tag{7}$$

In the formula: $\gamma$ is a constant, the general value in the liquid phase is 7; $\rho_0$ is the initial density, where the density of water is 998 kg/m$^3$. $B$ is used to limit the maximum change in density, taking $B = \frac{c_0^2\rho_0}{\gamma}$ in the fluid, $c_0$ is the sound speed.

In the simulation optimization step, a very important point is to determine the appropriate particle spacing and particle resolution (dp). Mintu considered various particle spaces from D/2 to D/6, and the results showed that the droplet trajectory tends to converge when the particle resolution gradually increases, but considering the cost of time and computational memory, the particle spacing of D/3 was found to be the best choice. Since variable particle spacing cannot be set in this algorithm, the validated D/3 optimal particle spacing is chosen for the simulation [33].

### 2.2. Coupling Wall Boundary Treatment

In this paper, the SPH-FEM coupling technique can be obtained in LS-DYNA to simulate the formation of sea spray, and is based on weakly compressible smooth particle hydrodynamics. The interaction between the FEM and the SPH particles was defined in the numerical simulation via the *CONTACT_NODES_TO_SURFACE contact card, with the master-slave penalty algorithm, in which the shell elements were assigned the role of

the master part and the SPH particles the slave part. In the numerical model applied in this paper, the surface of the hull structure is simplified as an immovable solid wall. Since the particles on or near the boundary are cut off by the boundary when integrating, because the particles on or near the boundary are only affected by the particles within the boundary, the particles must not penetrate the solid element during the interaction. Although the particle velocity on the boundary is zero, other variables are not necessarily zero, so this unilateral action may lead to failure of the solution, so the SPH particle algorithm is not suitable for the whole calculation area [37]. In order to solve this problem, the current mainstream methods are virtual particle method and penalty function method. The virtual particle method is unstable to generate virtual particles for a complex boundary, and it is easy to penetrate. The penalty function method is easy to implement, and it can deal with complex geometric boundary conditions [38].

In order to prevent the wave fluid particles from penetrating the hull structural unit, the following conditions need to be satisfied at the boundary:

$$\left(\frac{\partial}{\partial t}\mathbf{u} - \upsilon\right)\cdot\mathbf{n} = 0 \tag{8}$$

In the formula, $\mathbf{u}$, $\upsilon$ and $\mathbf{n}$ denote the displacements of element nodes at the boundary, the velocities of particles at the boundary and the normal vectors of elements at the boundary are represented, respectively.

In this paper, the coupling of SPH particles and FEM elements is solved by the contact algorithm based on the penalty function. The contact model is shown in Figure 1. For each fluid particle in the fluid domain, the adjacent finite element nodes are searched within a given smooth radius $H$, and all finite element surfaces belonging to FEM nodes are searched. When the SPH particle and the finite element surface satisfy the penetration depth $L$ of the particle to the finite element, it is considered that the particle collides. At this time, a penalty function is introduced, and a boundary contact force $\mathbf{f}_c$ is added to limit the penetration of fluid particles into the hull structure unit.

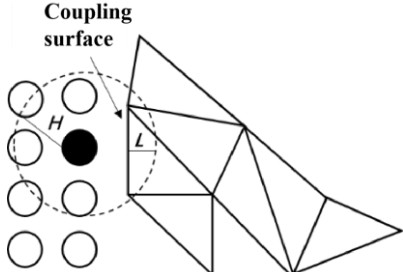

**Figure 1.** SPH-FEM Coupling surface.

When the hull structure is meshed, the finite element size and the smooth radius of the particles should not differ greatly, otherwise the finding omission phenomenon will occur. However, in order to reduce the memory of the mesh, and at the same time, the number of particles increases with the increase of the sea state level, etc., the missing phenomenon needs to be supplemented by the algorithm correction content. In other words, when the SPH particles at the boundary find the neighboring cells, the search is performed directly on the surface of the FEM cell [39]. Through the surface normal vector of each FEM cell and the position of the fluid particle at the boundary, the distance $r_{s-f}$ between the particle and the face of the cell is calculated, and then the contact force is calculated by the penalty function, and the contact force is applied to the fluid particle at the boundary as an external force. Since the finite element discretization of the solid is carried out by using tetrahedral cell in this paper, and each face of the tetrahedral cell is angular. Therefore, the process of detecting the contact between the particle and the cell surface can be converted into the process of detecting the triangular shape of the particle and the solid surface. The

boundary particle search process is shown in the following Figure 2. When particles cannot be retrieved around the mesh, which are defined as boundary particles.

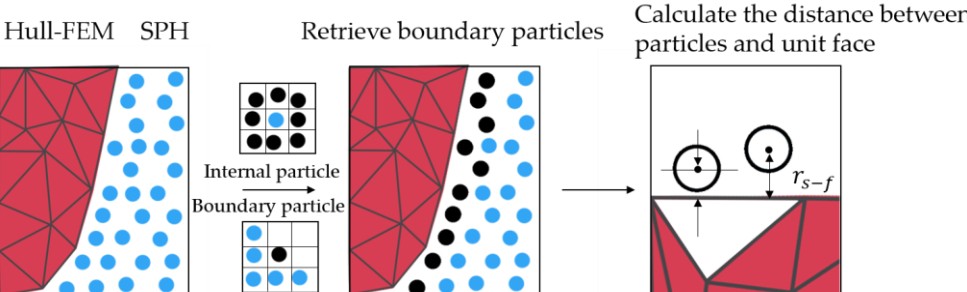

**Figure 2.** The relationship between boundary particles and cell surface.

The contact force at the boundary can be divided into normal contact force $\mathbf{f}_n$ and tangential contact force $\mathbf{f}_c$. The specific formula is as follows:

$$\mathbf{f}_n = -kl\mathbf{n} \tag{9}$$

$$l = h - r_{s-f} \tag{10}$$

$$\mathbf{f}_t = \eta\left(\frac{\partial u}{\partial t} - \mathbf{v}\right) \tag{11}$$

$$\mathbf{f}_c = \mathbf{f}_n + \mathbf{f}_t \tag{12}$$

where $k$ is the elastic coefficient, $\eta$ is the elastic potential energy density, $h$ is the smooth radius of the particle, and $n$ is the normal vector of the finite element surface.

Similarly, for the finite element, the contact force is also distributed to the FEM node as an external force for dynamic calculation, as shown in Equation (13):

$$\int_k^t = \frac{2w_k f_{con}}{1 + w_3^2 + w_3^2 + w_3^2}, \ k = \{1, 2, 3\} \tag{13}$$

where $w_k$ is the barycentric coordinate of the contact point between the fluid particle and the element surface relative to the triangular element surface. After the force on each node element is calculated, it can be summed to obtain the contact force on the whole solid node, which is applied to the solid together with gravity as an external force.

### 3. Method Validation Analysis

#### 3.1. Methods Analysis

Researchers have previously investigated the generation of sea spray from a single wave striking the hull of a ship. One of the more in-depth studies was conducted by Dehghani (2018) [40] for the examination of the trajectory study of sea spray. In order to determine the position, velocity, and acceleration of the sea spray, his paper makes the assumption that the sea spray is spherical and substitutes Newton's law based on the drag, body force, and added mass force on the sea spray. This study establishes the validation model in Dehghani's publication in order to determine whether the SPH algorithm suggested in this paper can be used as a method for sea spray particle calculation.

The arithmetic model in this paper is shown in Figure 3b, derived from the model of Dehghani et al. in Figure 3a. Since this paper only considers the distribution of droplets rising to descending to the hull as a percentage, the subsequent droplets moving with the wind on the hull are less (Dehghani, 2018), so the wind speed and the size of the droplets can be ignored. The total width of the hull is 7.3 m and the height of the foremast is 8 m, in order to measure the highest humidity of the foremast for comparison with the model calculated by Dehghani and Zakrzewski. The height of the superstructure is 4.5 m, which

is the same as the model for comparison, in order to obtain information on the height and position of the sea spray particles reaching the superstructure, and whether any sea spray particles can reach the top of the superstructure or even go beyond the superstructure to cover the rear part of the hull. The above three points are the relevant information points as the comparison. The calculated parameters are shown in Table 1.

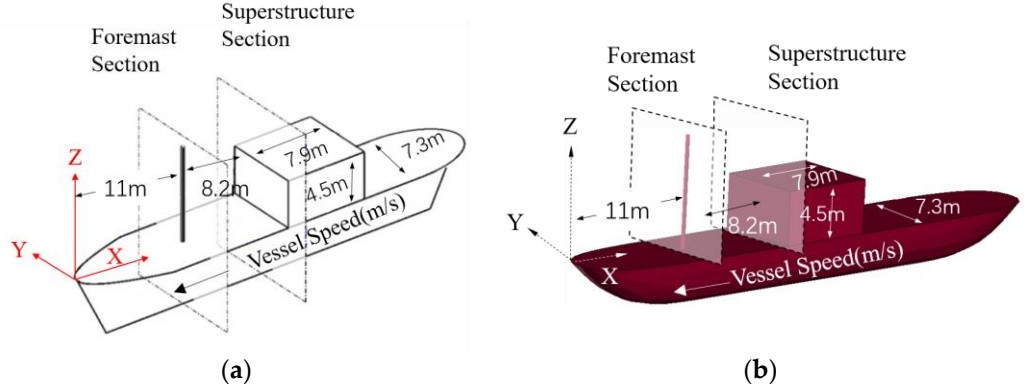

|  | (a) |  | (b) |

**Figure 3.** This figure shows the model built imitating the Dehghani model: (**a**) schematic of the MFV used as the ship model for the model of 3-D spray cloud by Dehghani (2018) [40], and (**b**) schematic of the MFV used as the ship model for the model of 3-D spray cloud by using SPH method.

**Table 1.** The calculation parameters of wave impacting hull forming ocean droplet model calculations.

| FEM (Hull) Parameters | Value | SPH Parameters | Value |
|---|---|---|---|
| Material density/(kg/m$^3$) | 7850 | Fluid particle density(kg/m$^3$) | 998 |
| Grid spacing/m | 0 | Initial spacing of fluid particles/m | 0.01 |
| Number of grids | 1067 | Number of fluid particles | 26,570 |
| Modulus of elasticity/MPa | $2.06 \times 10^5$ | Smooth kernel functions | Wendland |
| Poisson's ratio | 0.25 | Smooth length/m | 0.015 |
| Global damping factor | 0.00 | Pressure correction algorithm | CSPM algorithm |
| FEM Time step/s | $1.0 \times 10^{-4}$ | Viscosity coefficient | 0.3 |
|  |  | Sound velocity coefficient | 10 |
|  |  | Fluid dynamic viscosity/(Pa·s) | $1.0 \times 10^{-3}$ |
|  |  | SPH Time step/s | $1.0 \times 10^{-4}$ |

By using the coupled SPH-FEM algorithm for the numerical simulation of wave-induced sea spray, the process of wave impacting the hull and starting to break up to form sea spray is shown in Figure 4. Firstly, 26,570 water particles are generated as shown in the blue part of Figure 4a; through the motion, a wave of 3.45 m wave height is generated, at this time, the front part of the ship starts to contact with the wave current, which makes the wave current particles start to break up and initially form smaller sea spray as shown in Figure 4b. After the wave continues to impact the ship, the sea spray particles gradually increase and start to bifurcate, part of them first land on the ship, and the other part of the separated sea spray particles start to rise as shown in Figure 4c. A part of the separated sea spray particles formed a preliminary rising trend, this part of the particles were continuously impacted and began to rise as shown in Figure 4d. Sea spray particles with the impact force continued to rise to the highest point and began to land on the hull, the highest humidity in the front mast part of the hull, that is, the maximum height of the sea spray particles to reach the front mast is 6.84 m as shown in Figure 4e.

As can be seen from the simulation of flying particles in Figure 4a–e, wave particles begin to form flying particles after impacting the hull, part of the particles land on the hull, and part of the particles rise to the highest point. The SPH particles do not penetrate the FEM grid in this process, demonstrating the effectiveness of the penalty function contact algorithm used in this example. Secondly, to determine whether the coupling algorithm

used in this example is effective, it is possible to compare the stress time course at the point where SPH particles and FEM meet. Figure 5 fits the adjacent SPH and FEM pressure comparison graph; however, due to the number of SPH and the calculation method, the pressure rising and falling trend is slightly different from the value of the FEM grid by 0.1 s. As can be seen from the figure, there is no abrupt change point formed at the boundary, the pressure peaks of both systems are comparable, and the pressure values trend in the same direction. As a result, the coupling setting in this calculation can guarantee that the stress data from the near-zone SPH can be transmitted to the FEM unit.

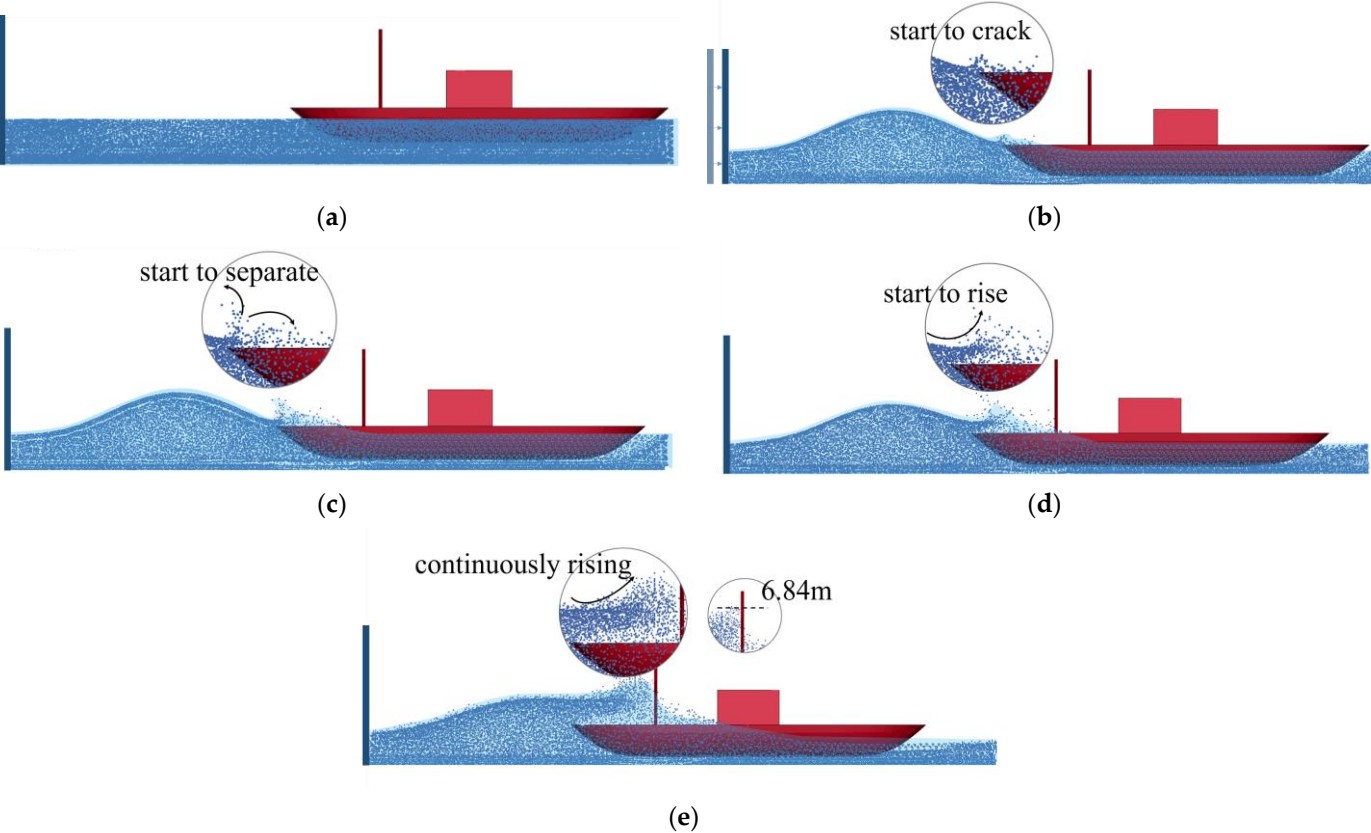

**Figure 4.** This is the sea spray generation process: (**a**) the ship is placed in the current particles, (**b**) waves begin to form sea spray, (**c**) sea spray particles start to disperse, (**d**) sea spray particles start to rise, and (**e**) sea spray particles continue to rise to the highest point.

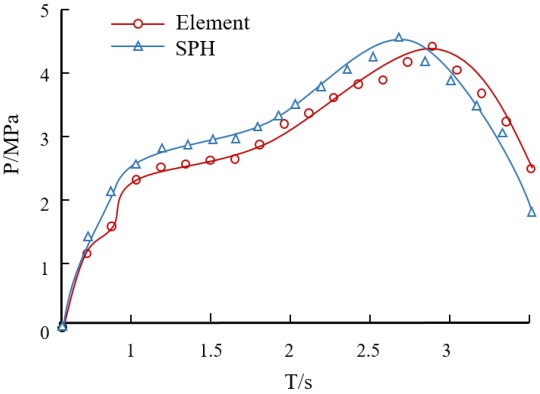

**Figure 5.** The pressure comparison curve between adjacent SPH particles and FEM grid.

### 3.2. Method Validation

The highest position of the sea spray particles in this mathematical example SPH model is depicted in Figure 6, according to a comparison of the maximum position of the sea spray trajectory in Dehghani's (2018) study. The greatest value of the sea spray from their model calculation is lower than the highest value from the SPH model because the data provided by Dehghani et al. include the LWC value and the wave height operating on the hull is 3.09 m. The graphs also show that the greatest values for the SPH model and for the Dehghani et al. model are both approximately 5–6 m on the ship's foredeck, and that the rising and decreasing trends of the sea spray particles are identical. In addition, at 11 m at the front mast, the highest humidity of Dehghani's 2-D model is at 6.28 m, the 3-D model is at 6.25 m, and the model of this calculation is at 6.84 m, which is within the range of the decreasing trend. The results of this computation are acceptable because the goal of this publication is to measure the number of occupied particles in a circumstance, which is distinct from the goal of Dehghani et al. study.

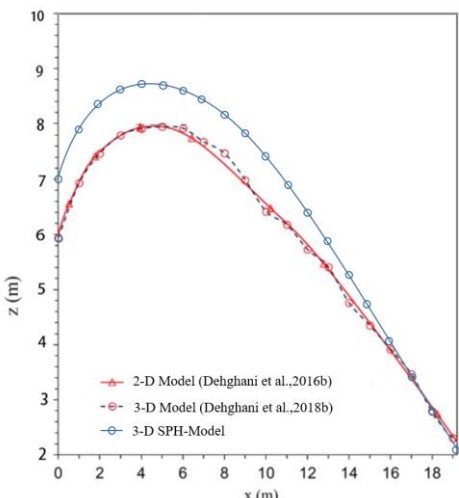

**Figure 6.** The maximum height of the model in this example is compared with the two-dimensional and three-dimensional models of Dehghani et al. (2016,2018) [40,41].

The calculation range can be broadly defined as 0–12 m and 12–18.5 m by the deck sea spray distribution map of Figure 7a. In the case of Dehghani et al., the water sea spray particles are split into particles of different sizes for normalized statistics. According to the literature model Figure 7b, the smaller water droplets are concentrated between 12 and 18.5 m from the bow, while the larger water droplets are concentrated in great numbers near the tip of the bow. SPH particles are denser at the bow's tip and less dense between 12 and 18.5 m away.

Finally, the comparison between the maximum humidity of the front mast and the maximum humidity of the front side of the superstructure is reported by the field observations of Zakrzewski (1988) et al. and supplemented by the analysis of Dehghani et al. Table 2 shows the reported data from previous studies and the numerical simulation results of the arithmetic example in this paper. According to the results of Figure 6, the front mast's maximum humidity is within the trend of droplet descent and the maximum humidity of the superstructure's front side is within a tolerable range when compared to the findings of earlier research. In contrast to the three earlier experiments, there is no droplet concentration on the top of the superstructure.

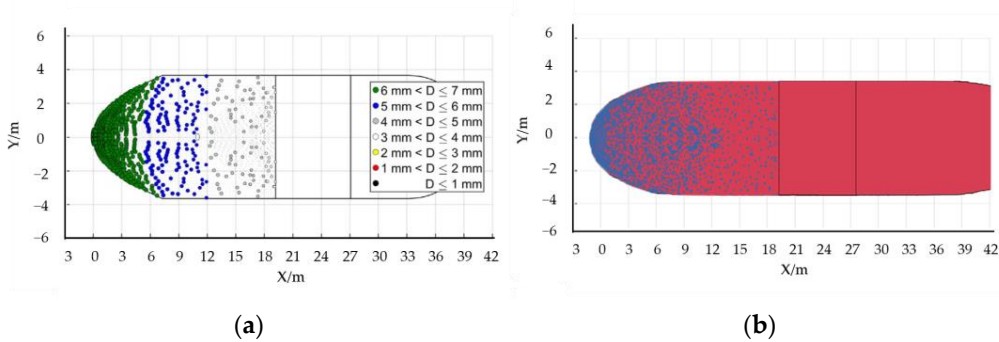

**Figure 7.** The deck droplet distribution for both models: (**a**) Dehghani et al. (2018) droplet distribution in the impact deck [40] and (**b**) SPH model arithmetic example of droplet distribution on impact deck.

**Table 2.** A comparison between the results of 2-D and 3-D models, field observations, and previous data.

| Positions on the Vessel | Result | | | |
|---|---|---|---|---|
| | Zakrzewski et al. (1988) [30] | Dehghani et al. 2-D (2016b) [41] | Dehghani et al. 3-D (2018b) [40] | SPH-Model |
| Wet height of the foremast | 5.85 m | 6.28 m | 6.25 m | 6.84 m |
| Front side of the superstructure | 2.07 m | 2.33 m | 2.6–3.5 m | 3.1 m |
| Roof of the superstructure | No spray | No spray | No spray | No spray |

## 4. Case and Analysis of Calculation Results

There is no report on the droplet distribution of oceanic droplets in the existing research on medium-sized container ships. This paper adopts the SR108 container ship [42], which already has the relevant ship motion performance study, as an example to do the model study. Since this paper only considers the wave droplet conditions of a single cycle, the stationary results at the end of a single cycle need to be obtained, ignoring the phenomenon of smaller droplets in the ship's continuous motion, so the model can ignore the influence of wind speed. Considering that the wave is the most direct factor causing the ocean droplet, and the wave height is small in the 1–3 level sea state, which is not close to the height of the ship, four different standard sea state schemes of level 4, level 5, level 6 and level 7 are selected to do the analysis of the wave impact hull model. Among them, Scheme 1 uses the fourth level sea state wave height, Scheme 2 uses the fifth level sea state wave height, Scheme 3 uses the sixth level sea state wave height, and Scheme 4 uses the seventh level sea state wave height. The length of the ship model is 175 m, the draft is 4.72 m, the width is 8.5 m, the speed of the ship is 24 m/s, and the incident wavelength is 175 m. Figure 8.

The results from the numerical simulation of wave impact on marine hull structures can be used as a reference for ice accumulation model predictions for ships sailing in polar regions. The flux percentage of ocean droplets on the hull is one of the primary reference factors for ice accretion, so the main objective of this research is to determine how many droplet particles remain on the hull following a single wave cycle that impacts it. The number of particles is calculated within one-third of the hull position from the bow, and the following scenario model is simulated using the algorithm settings already verified in Section 3.

One of the scenarios simulates the working condition of wave impacting the hull under the fourth level of sea state Figure 9. Under the realistic sea state operation, when the sea state enters the fourth level, the waves on the sea surface start to have a more obvious shape and will form waves of 1.25 to 2.5 m in height on the sea surface. The outcomes of the simulations determined by deriving the wave values demonstrate that the waves currently lack the capacity to produce sea spray for the ship hull used in the simulation.

There will be no impact on the ice buildup on the ship deck and superstructure because the water sea spray produced by the wave impacting the hull fall back to the surface before they can rise to the hull deck.

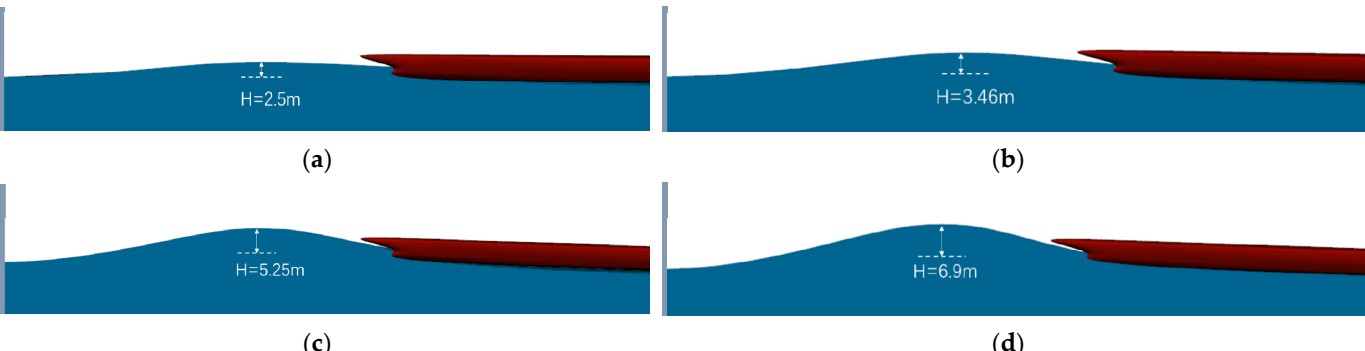

**Figure 8.** Four models with different levels of wave height: (**a**) four-level sea state simulation scheme, (**b**) five-level sea state simulation scheme, (**c**) six-level sea state simulation scheme, and (**d**) seven-level sea state simulation program.

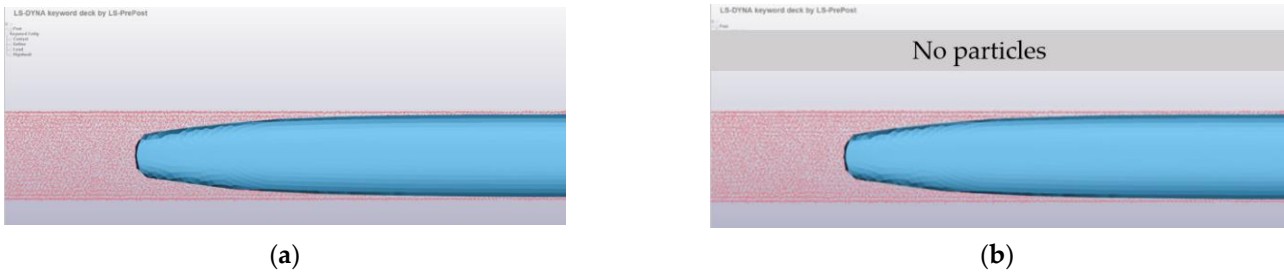

**Figure 9.** These are the two working conditions of Scenario 1: (**a**) conditions in the fourth-level sea state when waves hit the hull and (**b**) conditions when the fourth-level sea state's sea spray particles settle.

In scenario 2, waves impacting the hull under fifth-level sea conditions are simulated Figure 10. Since the impact on the hull structures at sea begins to increase when the sea state reaches the fifth level, it is often impossible to ignore the wave factor at this time. Larger wave crests start to appear on the sea surface at this level, and the waves take up a significant portion of the ocean and form waves that range in height from 2.5 to 4.0 m. The results of numerical simulation show that when the wave impacts the hull in class V sea state, a single wave has a number of 10,719 SPH particles, and sea spray start to form at the bow of the ship, and the droplets fall onto the deck after a period of operation. The number of ocean droplet particles produced when the wave hit the hull was 992, and the number of droplet particles after hitting the hull and falling into the hull was reduced by 412, showing that 41% of the droplets will return to the ocean surface quickly, and the remaining droplets will stay on the deck and are primarily concentrated on the sides of the deck, accounting for 9.25% of the individual waves.

Scenario 3 simulates the corresponding operating conditions in the sixth-level sea state Figure 11. Under the realistic sea state operation, when entering the sixth-level sea state, the wave crests on the sea surface sometimes take the shape of long waves of a storm, with tall crests everywhere, and the edges of the crests start to break and form huge waves. According to the simulation's findings, which were derived from wave height measurements, the waves in the class VI sea state contain 17,608 SPH particles. When the hull is struck by waves, a wider area of marine droplets has begun to form on the bow as a result of the wave contacting the hull, producing 1905 droplet particles. After the impact, there were still 1368 droplet particles on the hull's surface, which accounted for 10.81% of

the individual waves, and their layout range gradually shifted from the hull's sides to the center, eventually creating an arc.

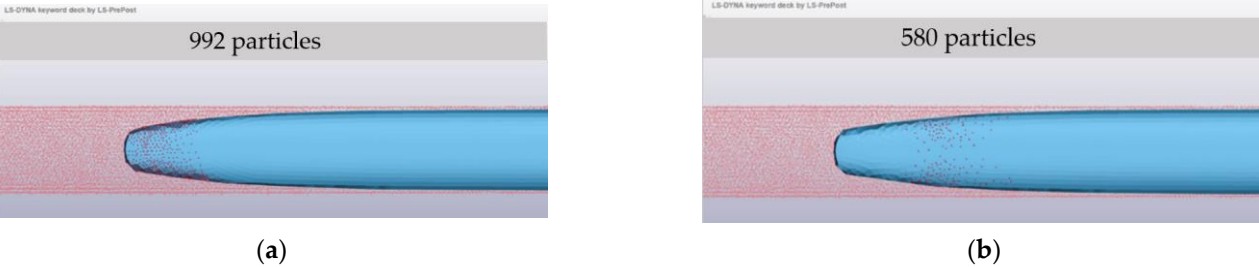

**Figure 10.** These are the two working conditions of Scenario 2: (**a**) conditions in the fifth-level sea state when waves hit the hull, and (**b**) conditions when the fifth-level sea state's sea spray particles settle.

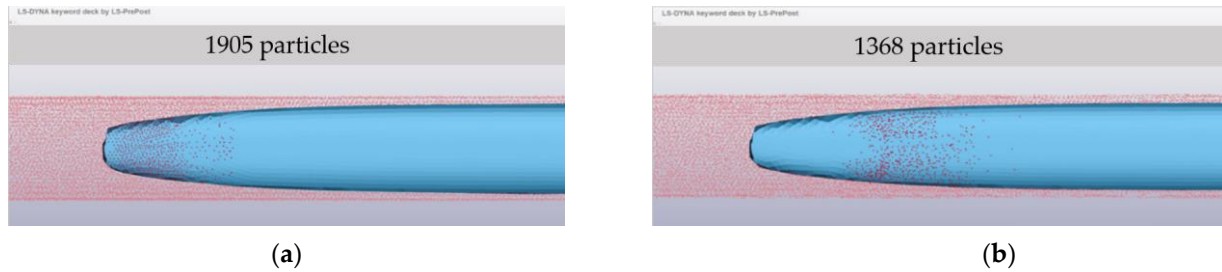

**Figure 11.** These are the two working conditions of Scenario 3: (**a**) conditions in the sixth-level sea state when waves hit the hull, and (**b**) conditions when the sixth-level sea state's sea spray particles settle.

The corresponding operating conditions about wave height under the seventh-stage sea state are simulated in Scenario 4 Figure 12. Under the realistic sea state operation, the sea surface begins to lurch, wave crests begin to roll, and flying foam can decrease visibility, which is a bad weather condition and makes navigation difficult for the steamer. According to the numerical simulation method, there are 21,238 SPH particles in the wave of the seventh -level sea state. The number of marine droplet particles created by the wave striking the hull is 2868, a significant increase from the above scheme. Additionally, there are 2153 droplet particles that fall on the hull, accounting for 13.51% of each wave, and the hull is completely covered in numerous droplets from both sides of the hull to the hull, with a large number of water particles and a clear arc-shaped distribution.

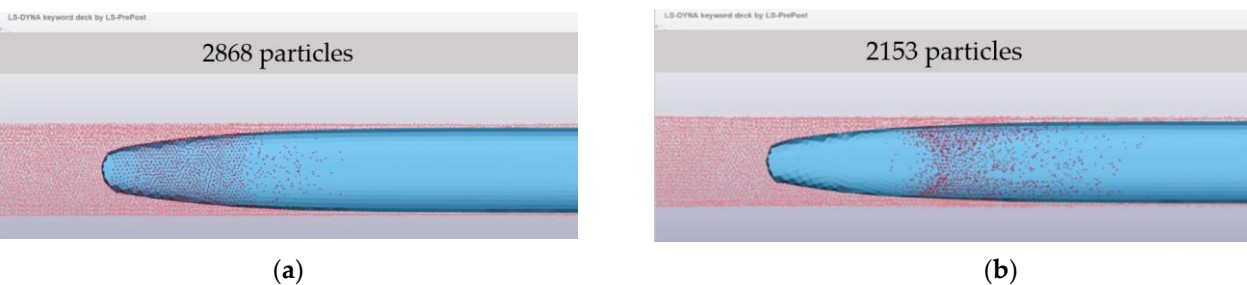

**Figure 12.** These are the two working conditions of Scheme 4: (**a**) conditions in the seventh-level sea state when waves hit the hull, and (**b**) conditions when the seventh-level sea state's sea spray particles settle.

## 5. Conclusions

From the numerical simulation and statistical results, we can analyze the relationship between the percentage of marine droplet particles on the hull according to the SPH

particles of individual waves as a reference, and can derive the percentage of marine sea spray particles on the hull under different sea conditions, respectively. In accordance with Figure 13a, the class V sea state will deposit 9.25% of the droplet particles in the hull, the class VI sea state will deposit 10.81% of the sea spray particles and the class VII sea state will deposit 13.51% of the sea spray particles. The ratio will rise as the wave value rises, and the relative particle number follows a similar pattern. At the same time, Figure 12b shows that when the droplet particles fall back to the hull, some particles will return to the sea through their own motion, and the trend of the number of particles on the hull is decreasing, however, the number of returning particles is more stable under different sea conditions, and as sea conditions increase, the change in the number of returning particles is smaller.

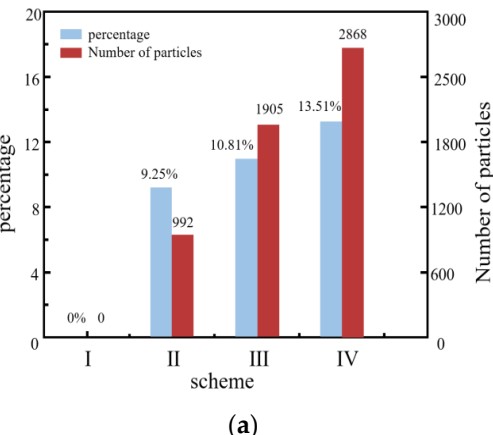
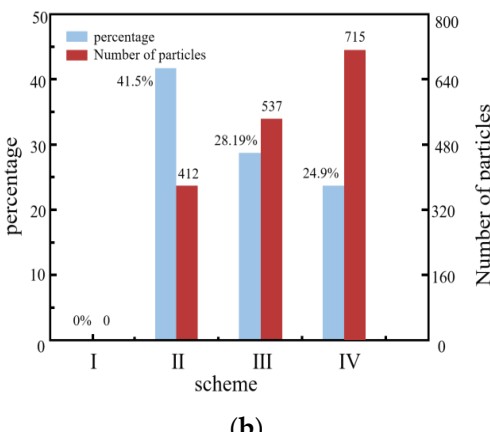

(**a**)　　　　　　　　　　　　　　　　(**b**)

**Figure 13.** (**a**) The number of particles on the hull and their share in the number of wave particles and (**b**) number of returning particles and their share in the number of particles in the hull.

In this paper, the SR108 container ship model, which already has the corresponding motion state research conclusions, is selected as the numerical simulation research model, and four different sea state simulation scenarios are set up to study the marine droplet working conditions formed by different wave heights impacting the hull in a single cycle, and further statistics are obtained on the effects of different sea state scenarios on the formation of marine droplets. The following conclusions can be drawn from the above study:

(1) After fifth-level sea state, a single cycle of waves will generate approximately 9.25% to 13.5% of the total number of particles on the hull of a single wave, and the higher the sea state level, the higher the number of particles and the higher the percentage of particles—where fifth-level sea state particles account for 9.25% of the total number of particles, six-level sea state accounts for 10.81%, and 13.51% of the seventh sea state.

(2) A small amount of sea spray particles returning to the sea surface soon after generation, and the particles that fall back to the sea surface are less related to the sea condition level and mainly influenced by the shape of the hull structure, which can be studied in more detail in the bow design for ice prevention later.

(3) The sea spray particles left on the hull are mainly distributed at both sides of the hull and gradually spread to the center as the sea state rises. When the sea state level is low, the attention to both sides of the hull needs to be increased when preventing icing, and when the sea state level is high, the attention to the ice accumulation on the superstructure in the center of the hull needs to be increased on the basis of the preliminary work.

(4) The coupled SPH-FEM method can simulate the distribution working conditions of marine droplets after interacting with the ship hull under different sea conditions, provide the physical modeling of droplet sources and effectively track the information of droplet particles, which can provide technical support for later prediction of droplet flux information related to the work of polar ship ice accumulation.

**Author Contributions:** J.C.; calculation method determined, put together the data, performed the data analysis and took part in the manuscript write-up. X.B.; contributed to the study conception and design, J.W.; collected the conference data and modeling, G.C. and T.Z.; simulation, and writing. All authors have read and agreed to the published version of the manuscript.

**Funding:** This research was supported by the National Natural Science Foundation of China (No. 51879125), the Postgraduate Research & Practice Innovation Program of Jiangsu Province (No. KYCX22_3777), the Natural Science Foundation of Jiangsu Province (No. BK20211342) and the Key R&D Projects in Guangdong Province (No. 2020B1111500001).

**Data Availability Statement:** The data presented in this study are available upon request from the corresponding author or the first author.

**Acknowledgments:** We thank the editor and anonymous reviewers for their comments, which considerably improved this work.

**Conflicts of Interest:** The authors declare no conflict of interest.

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
