# Peer review of "Research on Sea Spray Distribution of Marine Vessels Based on SPH-FEM Coupling Numerical Simulation Method"

_water, doi:10.3390/w14233834_

Round 1
Reviewer 1 Report
This article is still relatively complete and presents some preliminary ideas on the current polar field icing problem, which is enlightening and of some value. However, I think there are still several problems as follows:
Q1. The introduction mentions providing a reference for polar icing forecasting, and the context is polar icing, but there seems to be no brief overview of icing conditions, is the research direction of this paper relevant for icing forecasting?
Q2. The methodological overview of the existing research, there are various research methods of empirical equations, why not continue the existing empirical equations for calculation, and the advantages of this paper's method in comparison with them are not specified.
Q3. The paper proposes the droplet flux, but it does not explain the information of the physical quantity, such as what physical quantity the droplet flux is related to, so as to verify whether the research direction and ideas of this paper are reasonable.
Q4. The comparison picture of Figure2. is not clear, and the specific dimensions need to be clearly marked, because they are not mentioned in detail in the paper, which are easily confused.
Q5. The particles of the sea spray pictures after Figure8. are not clear enough, so if you can, you can consider whether you need to highlight them. Also the format of the acknowledgements and references is not correct and needs to be revised.
Reviewer 2 Report
In order to investigate the phenomenon of ice formation on ships, the authers use the SPH-FEM coupling approach , and track the data pertaining to the wave current particles by simulating the impact of a single wave on the ship hull under the wave height standard of various sea conditions. The research results have a certain guiding significance for the prediction of ship icing process.Here are some questions that the author needs to answer further. 1.What software tool is used by the author for calculation and simulation. 2.The mesh quality is very important for finite element calculation, and the author should discuss it. sunch as DOI: 10.1021/acs.energyfuels.0c03663;10.1021/acs.energyfuels.0c03979;10.1016/j.jngse.2021.103856. 3.A paragraph should be added in the introduction to explain the structure and innovation of the article.
